# DiLQR: Differentiable Iterative Linear Quadratic Regulator via Implicit Differentiation

Shuyuan Wang [1]   Philip D. Loewen [1]   Michael G. Forbes [2]   R. Bhushan Gopaluni [1]   Wei Pan [3]

## Abstract

While differentiable control has emerged as a powerful paradigm combining model-free flexibility with model-based efficiency, the iterative Linear Quadratic Regulator (iLQR) remains underexplored as a differentiable component. The scalability of differentiating through extended iterations and horizons poses significant challenges, hindering iLQR from being an effective differentiable controller. This paper introduces DiLQR, a framework that facilitates differentiation through iLQR, allowing it to serve as a trainable and differentiable module, either as or within a neural network. A novel aspect of this framework is the analytical solution that it provides for the gradient of an iLQR controller through implicit differentiation, which ensures a constant backward cost regardless of iteration, while producing an accurate gradient. We evaluate our framework on imitation tasks on famous control benchmarks. Our analytical method demonstrates superior computational performance, achieving up to **128x speedup** and a minimum of **21x speedup** compared to automatic differentiation. Our method also demonstrates superior learning performance ($\mathbf{10^6 x}$) compared to traditional neural network policies and better model loss with differentiable controllers that lack exact analytical gradients. Furthermore, we integrate our module into a larger network with visual inputs to demonstrate the capacity of our method for high-dimensional, fully end-to-end tasks. Codes can be found on the project homepage https://sites.google.com/view/dilqr/.

[1]The University of British Columbia, Vancouver, Canada [2]Honeywell Process Solutions, Vancouver, Canada [3]The University of Manchester, Manchester, England. Correspondence to: Wei Pan <wei.pan@manchester.ac.uk>, Bhushan Gopaluni <bhushan.gopaluni@ubc.ca>.

*Proceedings of the $42^{nd}$ International Conference on Machine Learning*, Vancouver, Canada. PMLR 267, 2025. Copyright 2025 by the author(s).

## 1. Introduction

Differentiable control has emerged as a powerful approach in the fields of reinforcement learning (RL) and imitation learning, enabling significant improvements in sample efficiency and performance. By integrating control policies into a differentiable framework, researchers can leverage gradient-based optimization techniques to directly optimize policy parameters. This integration allows for end-to-end training, where both the control strategy and the underlying model can be learned simultaneously, enhancing the adaptability and precision of control systems.

As a numerical controller, the iterative Linear Quadratic Regulator (iLQR) (Todorov et al., 2012) has been extensively adopted for trajectory optimization (Spielberg et al., 2021; Choi et al., 2023; Zhao et al., 2020; Mastalli et al., 2020) due to its computational efficiency (Tassa et al., 2014; Dean et al., 2020; Collins et al., 2021) and excellent control performance (Dantec et al., 2022; Xie et al., 2017; Chen et al., 2017). To make iLQR trainable as a neural network module, naively differentiating through an iLQR controller may be a reasonable choice, but the scalability of differentiating through hundreds of iterations steps poses a significant challenge, as the forward and backward passes during training are coupled. The forward pass involves iteratively solving an LQR optimization problem to converge on the optimal trajectory. The backward pass computes gradients through backpropagation, and becomes increasingly complex as it needs to traverse through all the layers of the forward pass, which requires significant computational resources (time and memory), especially for tasks requiring long iterations and long horizons. This coupling not only increases memory usage, but also significantly slows down the training process, making it difficult to scale to larger problems.

Efficient differentiable controllers are especially valuable in systems involving neural networks, such as multi-modal frameworks (Mao et al., 2023; Xu et al., 2024b; Xiao et al., 2022) and deep reinforcement learning (Ye et al., 2021; van Hasselt et al., 2016), where an upstream neural network module is required. Developing differentiable controllers with efficient gradient propagation is crucial, as they greatly enhance sample efficiency and reduce computational time for online tuning.

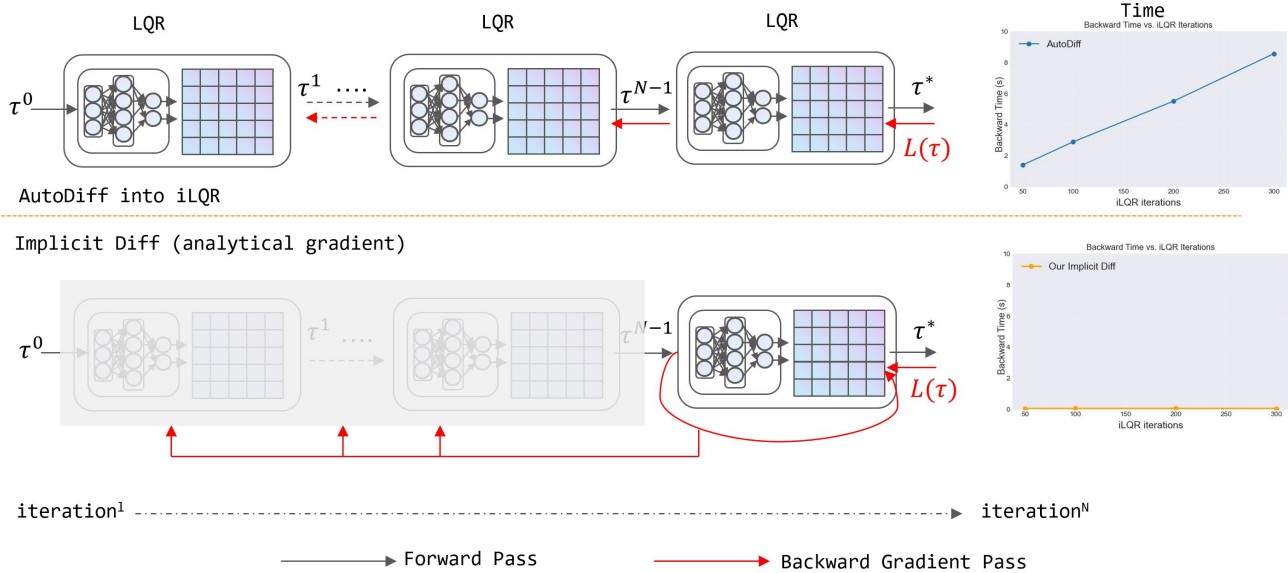

Figure 1: An overview of iLQR, and AutoDiff vs our proposed planner with implicit differentiation. As shown in the flowchart, autodiff must backpropagate through each layer of the LQR process, which leads to significantly increased memory usage to store intermediate gradients and computational load. In contrast, our proposed planner, using implicit differentiation, only needs to handle the final layer. This results in constant computational costs and memory usage, making our method much more efficient.

Developing analytical solutions would greatly alleviate these challenges. DiffMPC (Amos et al., 2018) pioneered the use of analytical gradients in LQR control, leading to significant improvements in computational efficiency and generalization of the learned controller. Its success has inspired extensions in various planning and control applications (East et al., 2020; Romero et al., 2024b; Karkus et al., 2023; Cheng et al., 2024; Soudbakhsh et al., 2023; Shrestha et al., 2023). Numerous studies have since shown that analytical gradients significantly improve learning performance, reducing computational costs, and improving scalability in complex, long-horizon tasks (Jin et al., 2020; Xu et al., 2024a; Jin et al., 2021; Böttcher et al., 2022; Zhao et al., 2022).

In this paper, we introduce an innovative analytical framework that leverages implicit differentiation to handle iLQR at its fixed point. This approach effectively separates the forward and backward computations, maintaining a constant computational load during the backward pass, irrespective of the iteration numbers for iLQR. By doing so, our method significantly reduces computational time and the memory usage needed for training, thereby enhancing scalability and efficiency in handling non-convex control problems. See Figure 1 for a conceptual comparison.

This paper makes the following contributions.

1. We develop an efficient method for analytical differentiation. We derive analytical trajectory derivatives for optimal control problems with tunable additive cost functions and constrained dynamics described by first-order difference equations, focusing on iLQR as the controller. Our analytical solution is exact, considering the entire iLQR graph. The method guarantees $O(1)$ computational complexity with respect to the number of iteration steps.

2. We propose a forward method for differentiating linearized dynamics with respect to nonlinear dynamics parameters, achieving speeds dozens of times faster than auto-differentiation tools such as `torch.autograd.jacobian`. Furthermore, we exploit the sparsity of the tensor expressions to compute some tensor derivatives that scale linearly with trajectory length.

3. We demonstrate the effectiveness of our framework in imitation and system identification tasks using the inverted pendulum and cartpole examples, showcasing superior sample efficiency and generalization compared to traditional neural network policies. Finally, we integrate our differentiable iLQR into a large network for end-to-end learning and control from pixels, demonstrating the extensibility and multimodal capabilities of our method.

**Notation**   For a function $f(\theta, g(\theta))$, where $g$ depends on $\theta$, we distinguish between **partial** and **total** derivatives. The **partial** derivative $\frac{\partial f}{\partial \theta}$ treats $g$ as independent, while the **total**

derivative accounts for indirect dependencies via the chain rule: $\frac{\mathrm{d}f}{\mathrm{d}\theta} = \frac{\partial f}{\partial \theta} + \frac{\partial f}{\partial g}\frac{\mathrm{d}g}{\mathrm{d}\theta}$. For readability, we use $\nabla_\theta f$ as a shorthand for $\frac{\mathrm{d}f}{\mathrm{d}\theta}$, emphasizing the total variation. When $f$ and $\theta$ are vectors or tensors, $\frac{\partial f}{\partial \theta}$ represents its Jacobian. See eq. (2) for concrete examples.

## 2. Related work on differentiable planning

Pure model-free techniques for policy search have demonstrated promising results in many domains by learning reactive policies that directly map observations to actions (Haarnoja et al., 2018; Sutton & Barto, 2018; Schulman et al., 2017; Fujimoto et al., 2018). However, due to the black box nature of these policies, model-free methods suffer from a lack of interpretability, poor generalization, and high sample complexity (Ye et al., 2021; Yu, 2018; Bacon et al., 2017; Deisenroth & Rasmussen, 2011). Differentiable planning integrates classical planning algorithms with modern deep learning techniques, enabling end-to-end training of models and policies, thereby combining the complementary advantages of model-free and model-based methods. Value Iteration Network (VIN) (Tamar et al., 2016) is a representative work that performs value iteration using convolution on lattice grids and has been extended further (Niu et al., 2018; Lee et al., 2018; Chaplot et al., 2021; Schleich et al., 2019). These works have demonstrated significant performance improvements on various tasks.

However, these works primarily focus on discrete action and state spaces. In the field of continuous control, most efforts have focused on differentiable LQR, including differentiating through finite horizon LQR (Amos et al., 2018; Shrestha et al., 2023), infinite horizon (East et al., 2020; Brewer, 1977), and constrained LQR (Bounou et al., 2023). References (Jin et al., 2020; 2021; Böttcher et al., 2022) propose frameworks that can differentiate through Pontryagin's Maximum Principle (PMP) conditions. However, the convergence speed of PMP-based methods is slower than that of iLQR (Jin et al., 2020), due to the 1.5 order convergence rate of iLQR. More importantly, these methods and their enhanced approach (Xu et al., 2024a) assume a broad range of forward pass solutions and do not align the gradient in the backward pass with the forward solution.

For iLQR, which is a powerful numerical control technique (Todorov et al., 2012; Li & Todorov, 2004; Zhu et al., 2023), (Tamar et al., 2017) differentiates through an iterative LQR (iLQR) solver to learn a cost-shaping term offline. Other methods based on numerical control techniques include (Okada et al., 2017; Pereira et al., 2018), which provide methods to differentiate through path integral optimal control, and (Srinivas et al., 2018), which shows how to embed differentiable planning (unrolled gradient descent over actions) within a goal-directed policy.

However, all of these methods require differentiation

through planning procedures by explicitly unrolling the optimization algorithm itself, introducing drawbacks such as increased memory and computational costs and reduced computational stability (Zhao et al., 2022; Bai et al., 2019). DiffMPC (Amos et al., 2018) is a representative work in the field of differentiable MPC. Significant progress has been made in the efficient differentiable LQR with box constraints by (Amos et al., 2018). To differentiate iLQR, (Amos et al., 2018) proposes a methodology that differentiates through the last layer of iLQR to avoid unrolling the entire iLQR graph. However, (Amos et al., 2018; Dinev et al., 2022) treats the input to the last layer of LQR as a constant, rather than a function of the learning parameters. Using implicit differentiation, we develop a framework that provides exact analytical solutions for iLQR gradients, improving the gradient computation presented in (Amos et al., 2018). Our approach not only addresses scalability issues, but also improves learning performance.

## 3. Background

The Iterative Linear Quadratic Regulator (iLQR) addresses the following control problem:

$$\min_{x_{1:T}, u_{1:T}} \sum_{t=1}^{T} g_t(x_t, u_t) \tag{1}$$
$$\text{s.t. } x_{t+1} = f_t(x_t, u_t), \ x_1 = x_{init}; \quad \underline{u} \leq u \leq \bar{u}.$$

At each iteration step, it linearizes the dynamics and makes a quadratic approximation of the cost function to produce a finite-time Linear Quadratic Regulator (LQR) problem. Solving this auxiliary problem produces updates for the original trajectory. Here are some details.

### 3.1. The Approximate Problem

Iteration $i$ begins with the trajectory $\tau^i = \{\tau_1^i, \ldots, \tau_T^i\}$, where $\tau_t^i = \{x_t^i, u_t^i\}$. For $t = 1, 2, \ldots, T$, we linearize the dynamics by defining

$$D_t = [A_t, B_t] = \left[ \frac{\partial f_t}{\partial x}\bigg|_{\tau_t^i}, \frac{\partial f_t}{\partial u}\bigg|_{\tau_t^i} \right]$$
$$d_t = f_t(x_t^i, u_t^i) - D_t \begin{bmatrix} x_t^i \\ u_t^i \end{bmatrix}, \tag{2}$$

and form a quadratic approximation of the cost function using

$$c_t^\top = [c_{t,x}, c_{t,u}] = \left[ \frac{\partial g_t}{\partial x}\bigg|_{\tau_t^i}, \frac{\partial g_t}{\partial u}\bigg|_{\tau_t^i} \right]$$
$$C_t = \begin{bmatrix} C_{t,xx} & C_{t,xu} \\ C_{t,ux} & C_{t,uu} \end{bmatrix}, \tag{3}$$

where

$$C_{t,xx} = \frac{\partial^2 g_t}{\partial x^2}\bigg|_{\tau_t^i}, \quad C_{t,uu} = \frac{\partial^2 g_t}{\partial u^2}\bigg|_{\tau_t^i}$$

$$C_{t,xu} = C_{t,ux}^\top = \frac{\partial^2 g_t}{\partial u \partial x}\bigg|_{\tau_t^i}.$$

These elements lead to an approximate problem whose unknowns are $\delta\tau_t = \tau_t - \tau_t^i$:

$$\min_{\delta\tau_{1:T}} \sum_{t=0}^T \frac{1}{2}\delta\tau_t^\top C_t \delta\tau_t + c_t^\top \delta\tau_t \tag{4}$$

$$\text{s.t. } \delta x_{t+1} = D_t \delta\tau_t, \ \delta x_1 = 0; \quad \underline{u} \le u \le \bar{u}.$$

### 3.2. The Trajectory Update

Problem (4) can be solved by the two-pass method detailed in (Tassa et al., 2014). First, a backward pass is conducted, using the Riccati-Mayne method (Mayne et al., 2000) to obtain a quadratic value function and a projected-Newton method to optimize the actions under box constraints. Then a forward pass uses the linear control gains $K_t, k_t$ obtained in the backward pass to roll out a new trajectory. Let $\delta\tau^\star$ denote the minimizing trajectory in (4). We use the controls in $\delta\tau^*$ directly, but discard the states in favor of an update based on the original dynamics, setting

$$u_t^{i+1} = u_t^i + \delta u_t^\star, \qquad x_{t+1}^{i+1} = f(x_t^{i+1}, u_t^{i+1}). \tag{5}$$

With these choices, defining $\tau_t^{i+1} = \{x_t^{i+1}, u_t^{i+1}\}$ provides a feasible trajectory for (1) that can serve as the starting point for another iteration.

## 4. Differentiable iLQR

### 4.1. End-to-end learning framework

In the learning problem of interest here, the cost functions $g_t$ and system dynamics $f_t$ involve structured uncertainty parameterized by a vector variable $\theta$. For example, in a drone, $\theta$ could represent physical parameters like mass or propeller length, while in a humanoid robot, it might refer to limb lengths or joint masses; additionally, $\theta$ can include reference trajectories for robot tracking, which help parametrize the cost function for control. Suppressing $\theta$ in the notation is typical when $\theta$ has a fixed value, but now we face the challenge of choosing $\theta$ to optimize some scalar criterion. This requires changing the notation to $f_t = f_t(x, u, \theta)$ and $g_t = g_t(x, u, \theta)$. As such, the derivatives shown in (2) and (3) must also be considered as functions of $\theta$. So, along a given reference trajectory $\tau$, the dynamics in (1) will generate three $\theta$-dependent matrices we must consider:

$$A_t(\theta) = \frac{\partial f_t}{\partial x}, \quad B_t(\theta) = \frac{\partial f_t}{\partial u}, \quad \text{and} \quad \frac{\partial f_t}{\partial \theta}.$$

The same is true for the coefficients in the quadratic approximation to the loss function in the original problem. Careful accounting for the $\theta$-dependence at every level is required for accurate gradients.

Suppose the loss function $L$ to be minimized by "learning" $\theta$ is expressed entirely in terms of the trajectory $\tau$. Then the influence of $\theta$ on the observed $L$-values will be indirect, and we will need the chain rule to express the gradient of the composite function $\theta \mapsto L(\tau(\theta))$:

$$\nabla_\theta(L \circ \tau)(\theta) = \nabla_\tau L(\tau(\theta))\frac{\partial \tau}{\partial \theta}. \tag{6}$$

In practical implementations, the partial derivatives required to form $\nabla_\tau L$ are provided during the backward pass by automatic differentiation tools (Paszke et al., 2019; Abadi et al., 2015). The main challenge, however, is to determine $\frac{\partial \tau}{\partial \theta}$, i.e., *the derivative of the optimal trajectory with respect to the learnable parameters*. This is the focus of the next section.

### 4.2. Fixed point differentiation

For a particular choice of $\theta$, we can consider the sequence of trajectories produced by iLQR:

$$\tau^0 \xrightarrow{LQR} \tau^1 \xrightarrow{LQR} \tau^2 \xrightarrow{LQR} \cdots \xrightarrow{LQR} \tau^\star \xrightarrow{LQR} \tau^\star \xrightarrow{LQR} . \tag{7}$$

Each iteration includes the three steps noted above: linearizing the system, conducting the backward pass, and performing the forward pass. Iterations proceed until the output $\tau^\star$ from an iLQR step is indistinguishable from the input, indicating that the process can no longer improve the input trajectory. This trajectory $\tau^\star$ is called a fixed point for the iLQR. We expect the value of $\theta$ to influence the fixed point produced above.

In general, an operator's fixed point can be calculated by various methods, typically iterative in nature. As pointed out in (Bai et al., 2019), naively differentiating through such a scheme would require intensive memory usage (Tamar et al., 2016; Lee et al., 2018) and computational effort (Zhao et al., 2022). Instead, we propose to use implicit differentiation directly on the defining identity. This gives direct access to the derivatives required by decoupling the forward (fixed-point iteration as the solver) and backward passes (differentiating through the solver).

Let us write $X = (x_1, \ldots, x_T)$ and $U = (u_1, \ldots, u_T)$ for the components of a trajectory $\tau = (x_1, u_1, x_2, u_2, \ldots, x_T, u_T)$, and abuse notation somewhat by identifying $\tau$ with $(X, U)$. At a fixed point $(X^\star, U^\star)$ of the iLQR process for parameter $\theta$, we have the following:

$$X^\star = F(X^\star, U^\star, \theta), \quad U^\star = G(X^\star, U^\star, \theta) \tag{8}$$

where $F$ and $G$ summarize the operations that define a single

iteration in the iLQR algorithm. (Thus eq. (8) formalizes the graphical summary in eq. (7).)

In eq. (8), the solutions $X^\star$ and $U^\star$ depend on the parameter $\theta$. By treating both $X^\star$ and $U^\star$ explicitly as functions of $\theta$, we can interpret eq. (8) as an identity valid for all $\theta$. Differentiating through this identity yields a new one:

$$\nabla_\theta X^\star = \frac{\partial F}{\partial X}\nabla_\theta X^\star + \frac{\partial F}{\partial U}\nabla_\theta U^\star + \frac{\partial F}{\partial \theta},$$
$$\nabla_\theta U^\star = \frac{\partial G}{\partial X}\nabla_\theta X^\star + \frac{\partial G}{\partial U}\nabla_\theta U^\star + \frac{\partial G}{\partial \theta}. \tag{9}$$

Here, the matrix-valued partial derivatives of $F$ and $G$ above are evaluated at $(X^\star(\theta), U^\star(\theta), \theta)$. Likewise, $\nabla_\theta X^\star$ and $\nabla_\theta U^\star$ are the Jacobians (sensitivity matrices) that quantify the $\theta$-dependence of the optimal trajectory; both depend on $\theta$. Rearranging eq. (9) produces a system of linear equations in which these two matrices provide the unknowns:

$$(I - \frac{\partial F}{\partial X})\nabla_\theta X^\star - \frac{\partial F}{\partial U}\nabla_\theta U^\star = \frac{\partial F}{\partial \theta},$$
$$-\frac{\partial G}{\partial X}\nabla_\theta X^\star + (I - \frac{\partial G}{\partial U})\nabla_\theta U^\star = \frac{\partial G}{\partial \theta}. \tag{10}$$

The analytical solution for this system is given below.

**Proposition 4.1.** *The Jacobians in eq. (10) are given by*

$$\nabla_\theta X^\star = M(F_\theta + F_U(K - G_X M F_U)^{-1}(G_X M F_\theta - G_\theta))$$
$$\nabla_\theta U^\star = (K - G_X M F_U)^{-1}(G_X M F_\theta + G_\theta), \tag{11}$$

*where we denote $M = (I - F_X)^{-1}$ and $K = I - G_U$, and use the condensed notation*

$$F_X = \frac{\partial F}{\partial X}, \quad F_U = \frac{\partial F}{\partial U}, \quad F_\theta = \frac{\partial F}{\partial \theta},$$
$$G_X = \frac{\partial G}{\partial X}, \quad G_U = \frac{\partial G}{\partial U}, \quad G_\theta = \frac{\partial G}{\partial \theta}. \tag{12}$$

See the Appendix A.1.

To be completely explicit, suppose a parameter $\theta$ is given. Then eq. (8) defines a fixed point $\tau^\star$ in terms of this particular $\theta$, and this $\tau^\star$ provides the evaluation point $(X^\star(\theta), U^\star(\theta), \theta)$ for all the Jacobian matrices involving $F$ and $G$ in Equations (9) to (11).

### 4.3. Obtaining each term

The functions $F$ and $G$ whose Jacobians appear in eq. (12) are defined by rather complicated arg min operations. The Chain-Rule pattern below, which we can apply to either $H = F$ or $H = G$, suggests that

$$H_X = \frac{\partial H}{\partial D}\frac{\partial D}{\partial X} + \frac{\partial H}{\partial d}\frac{\partial d}{\partial X} + \frac{\partial H}{\partial C}\frac{\partial C}{\partial X} + \frac{\partial H}{\partial c}\frac{\partial c}{\partial X},$$
$$H_U = \frac{\partial H}{\partial D}\frac{\partial D}{\partial U} + \frac{\partial H}{\partial d}\frac{\partial d}{\partial U} + \frac{\partial H}{\partial C}\frac{\partial C}{\partial U} + \frac{\partial H}{\partial c}\frac{\partial c}{\partial U}, \tag{13}$$
$$H_\theta = \frac{\partial H}{\partial D}\frac{\partial D}{\partial \theta} + \frac{\partial H}{\partial d}\frac{\partial d}{\partial \theta} + \frac{\partial H}{\partial C}\frac{\partial C}{\partial \theta} + \frac{\partial H}{\partial c}\frac{\partial c}{\partial \theta}.$$

In each term on the right, the first matrix factor (e.g., $\partial H/\partial D$) expresses the sensitivity of the optimal LQR trajectory with respect to the corresponding named ingredient of the formulation in eq. (4). Efficient methods for calculating these terms are known: see (Amos et al., 2018; Amos & Kolter, 2017). The second factor in each term of (13) can be computed using automatic differentiation. The next subsections detail how to calculate these terms efficiently.

### 4.4. Parallelization

(Amos et al., 2018) propose a method that directly calculates $\frac{\partial L}{\partial D}$, $\frac{\partial L}{\partial d}$, $\frac{\partial L}{\partial C}$, and $\frac{\partial L}{\partial c}$ with a complexity of only $O(T)$. We adopt these results in our framework. To facilitate parallelization, we construct batches of binary loss functions. Specifically, to compute $\frac{\partial H_{i,j}}{\partial D}$, we set the $L_{i,j}$ element in $L$ to 1, while all other elements are set to 0, and then calculate $\frac{\partial L}{\partial D}$. Although this approach introduces more computations, the computations can be fully parallelized since each operation is completely independent. As a result, the calculation of $\frac{\partial H}{\partial D}$ can be parallelized efficiently. The same method also applies to $\frac{\partial H}{\partial d}$, $\frac{\partial H}{\partial C}$, and $\frac{\partial H}{\partial c}$.

### 4.5. Exploiting sparsity

Some care is required when coding the calculations for which eq. (13) provides the models. With $X = (x_1, \ldots, x_T)$ as above, and the corresponding $D = (D_1, \ldots, D_T)$, the quantity $\frac{\partial D}{\partial X}$ suggests a huge structure involving $T^2$ submatrices of the general form $\frac{\partial D_t}{\partial x_{t'}}$. However, the definitions in eq. (2) show that any such submatrix in which $t' \neq t$ will be zero. Thus the matrix $\frac{\partial D}{\partial X}$ shown above is block diagonal. Our implementation never instantiates it. Instead, we work directly with the information-bearing diagonal blocks $\frac{\partial D_t}{\partial x_t}$, $1 \leq t \leq T$.

### 4.6. Forward algorithm

It can be costly to evaluate matrices like $\frac{\partial D}{\partial \theta}$. In Pytorch, for example, such tools such as `torch.autograd.jacobian` rely on backpropagation, which means that gradient information from one time step is not reused for the next time step. However, the derivation above makes it clear that knowing $\frac{\partial D_{t-1}}{\partial \theta}$ allows for a direct calculation of $\frac{\partial D_t}{\partial \theta}$.

We now propose an efficient **forward approach** that uses available information efficiently to accelerate later steps. We refer to $\frac{\partial D_t}{\partial \theta}$ from (13) as $\nabla_\theta D_t$ here for clarity and to distinguish it from other gradient notations, a convention we apply similarly to other gradients such as $\frac{\partial d_t}{\partial \theta}$.

Given a trajectory satisfying $x_{t+1} = f_t(x_t, u_t, \theta)$, the matrices $D_t$ and $d_t$ defined in eq. (2) are functions of $x_t$, $u_t$,

and $\theta$. For time step $t$, we will have

$$\nabla_\theta D_t = \frac{\partial D_t}{\partial \theta} + \left[ \frac{\partial D_t}{\partial x_t} + \frac{\partial D_t}{\partial u_t} \frac{\partial u_t}{\partial x_t} \right] \nabla_\theta x_t \qquad (14)$$

with

$$\nabla_\theta x_t = \frac{\partial x_t}{\partial \theta} + \left[ \frac{\partial x_t}{\partial x_{t-1}} + \frac{\partial x_t}{\partial u_{t-1}} \frac{\partial u_{t-1}}{\partial x_{t-1}} \right] \nabla_\theta x_{t-1}, \quad (15)$$

where $\frac{\partial D_t}{\partial \theta}$, $\frac{\partial D_t}{\partial x_t}$, $\frac{\partial D_t}{\partial u_t}$ and $\frac{\partial x_t}{\partial \theta}$, $\frac{\partial x_t}{\partial x_{t-1}}$, $\frac{\partial x_t}{\partial u_{t-1}}$ are analytically calculated in first so that on each time step we only need to instantly plug in the corresponding parameter values to obtain the numerical gradients. $\frac{\partial u_t}{\partial x_t}$ and $\frac{\partial u_{t-1}}{\partial x_{t-1}}$ are the linear control gain solved from FT-LQR. $\nabla_\theta x_{t-1}$ is the stored information from time step $t-1$ and reused here, and $\nabla_\theta x_t$ is prepared for the next time step $t+1$. Finally

$$\nabla_\theta d_t = \nabla_\theta x_{t+1} - \frac{\partial D_t}{\partial \theta} \begin{bmatrix} x_t \\ u_t \end{bmatrix} - D_t \begin{bmatrix} I \\ \frac{\partial u_t}{\partial x_t} \end{bmatrix} \nabla_\theta x_t,$$

$$\nabla_{x_t} d_t = -\frac{\partial D_t}{\partial x_t} \begin{bmatrix} x_t \\ u_t \end{bmatrix}, \quad \nabla_{u_t} d_t = -\frac{\partial D_t}{\partial u_t} \begin{bmatrix} x_t \\ u_t \end{bmatrix}. \qquad (16)$$

The calculation of $\nabla_\theta C_t$ and $\nabla_\theta c_t$ is similar.

---

**Algorithm 1** Forward Algorithm

---

1: **Input:** $\frac{\partial D_t}{\partial \theta}$, $\frac{\partial D_t}{\partial x_t}$, $\frac{\partial D_t}{\partial u_t}$ and $\frac{\partial x_t}{\partial \theta}$, $D_t$
2: Initialize variables $\nabla_\theta x_0 = 0$
3: **for** time step $t = 1, 2, \dots, T$ **do**
4:      obtain $\nabla_\theta x_t$ through (15)
5:      obtain $\nabla_\theta D_t$ with $\nabla_\theta x_t$ and (14), and obtain $\nabla_\theta d_t$ with $\nabla_\theta x_t$ and (16)
6: **end for**
7: **return** $\nabla_\theta D, \nabla_\theta d$

---

### 4.7. Methodological Comparison and Discussion

**Differences between our method and DiffMPC (Amos et al., 2018)** DiffMPC treats input $X^*$ and $U^*$ as constant and uses auto-differentiation to obtain $\frac{\partial D}{\partial \theta}$, and finally use the chain rule to obtain the derivative of the optimal trajectory. We improve DiffMPC by further considering the input $X^*$ and $U^*$ as a function of $\theta$, that is, $X^*(\theta)$ and $U^*(\theta)$, and leverage implicit differentiation on the fixed-point to *solve* the exact analytical gradient, improving the accuracy of the gradient. The box in 17 illustrates the differences between the two approaches

$$A^i(\tau^i, \theta) = \frac{\partial f(x, u, \theta)}{\partial x} \Bigg|_{\tau^i}, \quad \nabla_\theta A^i = \frac{\partial A^i}{\partial \theta} + \boxed{\frac{\partial A^i}{\partial \tau^i} \frac{\partial \tau^i}{\partial \theta}}. \tag{17}$$

## 5. Experiments

We follow the examples and experimental setups from previous works (Amos et al., 2018; Jin et al., 2020; Xu et al., 2024a; Watter et al., 2015) and conduct experiments on two well-known control benchmarks: CartPole and Inverted Pendulum. The experiments demonstrate our method's computational performance (at most **128x speedup**) and superior learning performance ($10^6$ improvement). All experiments were carried out on a platform with an AMD 3700X 3.6GHz CPU, 16GB RAM, and an RTX3080 GPU with 10GB VRAM. The experiments are implemented with Pytorch (Paszke et al., 2019).

### 5.1. Computational Performance

The performance of our differentiable iLQR solver is shown in Figure 2. We compare it to the naive approach, where the gradients are computed by differentiating through the entire unrolled chain of iLQR. The results of the experiments clearly demonstrate the significant computational advantage of our method over AutoDiff across all configurations.

**Backward pass efficiency:** For example, for a horizon of 10 and 300 iterations, AutoDiff takes 8.57 seconds compared to just 0.067 seconds with our method, resulting in a **128x speedup**. Even in the case with the smallest improvement—horizon of 10 and 50 iterations, AutoDiff takes 1.41 seconds, while our method remains 0.067 seconds, still delivering a **21x speedup**. These results highlight the clear scalability and efficiency of our method, maintaining a near-constant computation time as the number of iLQR iterations increases, while AutoDiff's time grows significantly with longer horizons and more iterations.

### 5.2. Imitation Learning

Imitation learning recovers the cost and dynamics of a controller through ***only actions***. Similarly to (Amos et al., 2018), we compare our approach with **Neural Network (NN)**: An LSTM-based approach that takes the state $x$ as input and predicts the nominal action sequence, directly optimizing the imitation loss directly; **SysId**: Assumes that the cost of the controller is known and approximates the parameters of the dynamics by optimizing the next-state transitions; and DiffMPC (Amos et al., 2018). We evaluated two variations of our method: DiLQR.dx: Assumes that the cost of the controller is known and approximates the parameters of the dynamics by directly optimizing the imitation loss; DiLQR.cost: Assumes that the dynamics of the controller are known and approximates the cost by directly optimizing the imitation loss. Following DiffMPC (Amos et al., 2018), we adopt the same dataset size convention where "train=50" and "train=100" denote the number of expert trajectories available during training.

**Imitation Loss:** In Figure 3, we compare our method with NN and Sysid using imitation loss, under the train=100 set-

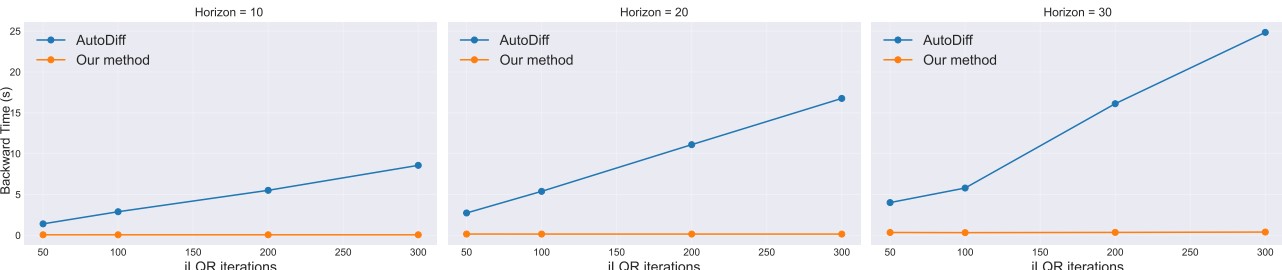

Figure 2: Backward computation time comparison between AutoDiff and our proposed method across different iLQR iterations and LQR horizons. AutoDiff's computation time scales linearly with the number of iterations, while our method maintains constant computation time. The experiments are conducted under pendulum domain, with batch size 20.

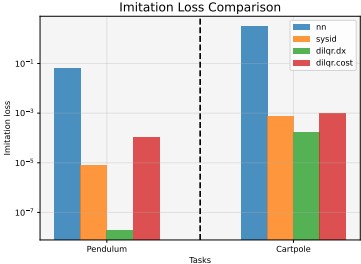

Figure 3: Learning results on the pendulum and cartpole. We select the best validation loss during training and report the test loss (Amos et al., 2018), averaged over five trials.

ting. Notably, our method performs the best in the dx mode across both tasks, achieving a performance improvement of orders of magnitude—$10^6$ and $10^4$—over the NN. In the dcost mode, our method is also dozens of times stronger than the NN but slightly weaker than Sysid. This is because Sysid directly leverages a system model with state estimates, while imitation learning relies solely on action data, which contains less information. The fact that our method achieves comparable results to Sysid in this mode demonstrates its effectiveness.

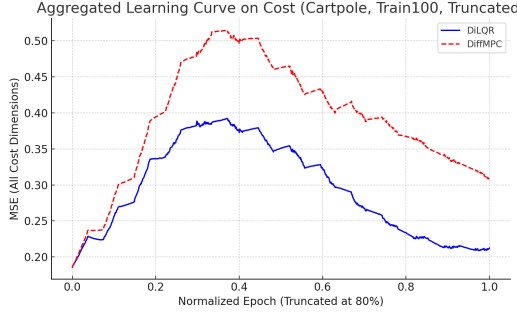

Figure 4: Model loss curves for cost function learning under the cartpole domain, trained for 500 epochs.

**Model Loss:** In Figure 4, we compare the model error learned from our approach to that of DiffMPC. Model loss is defined as the MSE($\theta - \hat{\theta}$), where $\theta$ represents the parameters of the cost function. In the dcost mode, our approach recovers more accurate model parameters than DiffMPC, reducing model loss by 32%, indicating an improvement over our analytical results.

We further evaluate the dynamics learning mode. Model loss here is defined as the MSE($\theta - \hat{\theta}$), where $\theta$ represents the parameters of the dynamics. Figure 5 shows DiffMPC's error plateaus early, while DiLQR achieves 41% lower final error through steady optimization (with train=50). Although DiLQR converges slightly slower with train=100, it maintains strong physical consistency (2.76% negative parameters vs. DiffMPC's 16.85%, Table 1), largely avoiding non-physical solutions that could destabilize control. These results demonstrate DiLQR's dual advantages in both accuracy and physical plausibility. See Appendix A.5 for clarification on the slight discrepancy with the rebuttal.

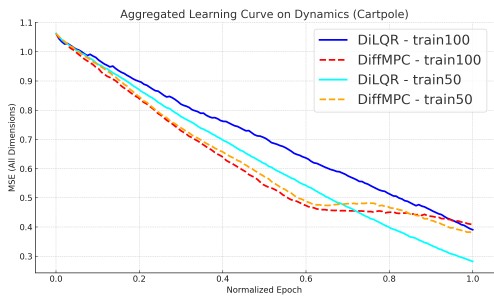

Figure 5: Model loss curves for dynamics learning under the cartpole domain, trained for 500 epochs.

Table 1: Bad-Value Ratio (% of negative values in learned dynamics)

| Train Size | DiLQR | DiffMPC |
|---|---|---|
| dx (train=100) | **2.76%** | 16.85% |
| dx (train=50) | 7.23% | 17.82% |

**Comparison to Other Differentiable Control Methods**
While SafePDP(Jin et al., 2021) and IDOC(Xu et al., 2024a) report an imitation loss of ∼1e-2 under its official codebase, these results are obtained under experimental settings that differ from ours—specifically, using ground-truth-near parameter initialization and expert trajectories generated by an oracle solver (IPOPT). We progressively align these conditions with our own experimental setup. As shown in Figure 6, once the expert trajectories are replaced with those generated by iLQR, DiLQR exhibits a clear performance advantage over both SafePDP and IDOC (As single-trajectory training is the default setting in both SafePDP and IDOC, all methods are trained using **the same single** expert trajectory generated by iLQR.). This highlights the methodological gap and underscores DiLQR's effectiveness for learning structured models in iLQR-specific settings.

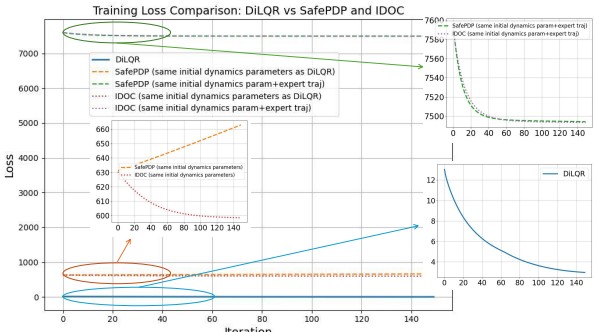

Figure 6: Training loss comparison on the Cartpole task. DiLQR is compared against SafePDP and IDOC under two settings: (1) same initial dynamics parameters as DiLQR, and (2) same dynamics parameters plus same expert trajectories. Losses are plotted as reported by each method. DiLQR demonstrates significantly lower final loss.

## 5.3. Ablation study

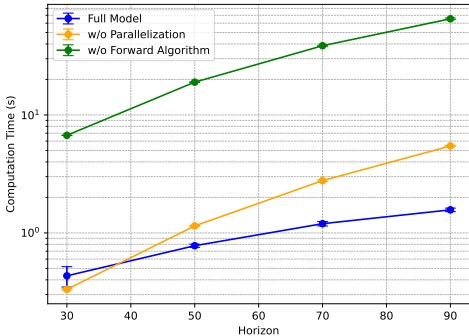

Figure 7: Computation time comparison across different ablation settings as a function of horizon length. Error bars measure standard error across 5 samples.

To quantify the contribution of each module, we compare the computation time of different ablation configurations, including removing parallelization (Sec. 4.4), sparsity exploration (Sec. 4.5), and the forward algorithm (Sec. 4.6). We evaluate these configurations on Pendulum across varying horizons.

**Computational Time** Parallelization (Sec. 4.4) and the forward algorithm (Sec. 4.6) significantly reduce computation time. As shown in Figure 7, removing the forward algorithm leads to a **sharp increase** in computation time across all horizons, while disabling the parallelization further exacerbates this effect, especially for longer horizons.

**Discussion** Among the tested modules, **the forward algorithm (Sec. 4.6) has the greatest impact on computation time**, especially for long-horizon settings. Parallelization also plays a critical role in reducing computational cost. These results validate our design choices and highlight the necessity of each component for scalable differentiable control. Interestingly, at horizon 30, the computation without parallelization is slightly faster than with parallelization. This is due to the lower clock frequency of our GPU compared to the CPU. In practice, an adaptive switching scheme would be employed to select the optimal computation strategy based on the horizon and batch size.

## 5.4. Visual control

We next explore a more complex, high-dimensional task: controlling an inverted pendulum system using images as input.

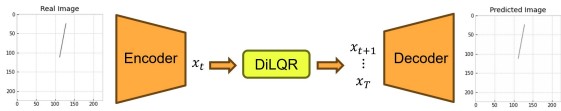

Figure 8: Diagram of the end-to-end control architecture. The encoder maps compressed set of four input frames to the physical state variables. The differentiable iLQR then steps the state forward using the encoder's parameters. The decoder takes the predicted state and generates a future frame to match true future observation.

In this task, the state of the pendulum is visualized by a rendered line starting from the center of the image, with the angle representing the position of the pendulum. The objective is to swing up the underactuated pendulum from its downward resting position and balance it. The network architecture consists of a mirrored encoder-decoder structure, each with five convolutional or transposed convolutional layers, respectively. To capture the velocity information, we stack four compressed images as input channels. An example of observations and reconstructions is provided in Figure 8.

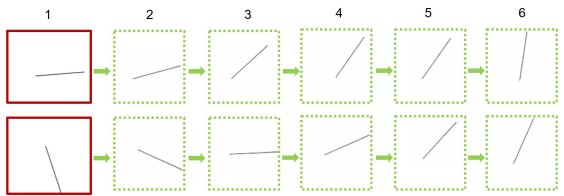

Figure 9: Imagined trajectory in the pendulum domain. The first image (red) represents the real input, while the following images are "dreamed up" by our model based on the initial image.

Our modular approach handles the coordination between the controller and decoder seamlessly. Figure 9 shows sample images drawn from the task depicting a trajectory generated by our system. In this scenario, the system is given just one real image and, with the help of DiLQR, it can output a sequence of predicted images, which closely approximate the actual trajectory of the pendulum. We further design an autoencoder that connects the trained encoder and decoder to predict trajectory images in an autoregressive manner. As shown in Figure 10, the network incorporating differentiable iLQR achieves significantly higher prediction accuracy with lower variance.

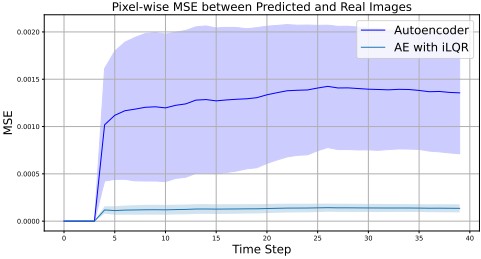

Figure 10: Comparison of image prediction error across trajectory steps. The shaded area denotes the standard error over 25 trials.

# 6. Discussion

## 6.1. Scope of Experiments

In this paper, we focus on the theoretical aspects of differentiable control methods. While our experiments are based on simpler control tasks, the advantages of our approach promise to extend to more complex, real-world applications. Many prior works (Amos et al., 2018; Watter et al., 2015; Xu et al., 2024a; Jin et al., 2020) also rely on such toy examples to demonstrate foundational concepts. A demonstration of our method on a high-dimensional rocket control task is available on the project page.

## 6.2. Future direction

One promising direction is embedding our differentiable controller into RL frameworks. For instance, it could be inte-

grated into a policy network and trained using an actor-critic approach, enabling more efficient policy updates (Romero et al., 2024a; Romero et al.). With its ability to propagate gradients through the control process, our method could enhance RL's performance, potentially achieving state-of-the-art results in more advanced tasks.

Another direction is combining our method with perceptual control frameworks such as DiffTORI (Wan et al., 2024). While DiffTORI emphasizes generality and visual input handling, our approach offers precise and efficient gradients for structured systems. Integrating the two could enable scalable, end-to-end learning in high-dimensional, multi-modal tasks.

## 6.3. Comparison with General Differentiation Frameworks.

Our method addresses a key limitation in general-purpose frameworks such as Theseus (Pineda et al., 2022) or JAXopt, which can differentiate through fixed-point equations of the form $x^* = f(x^*)$. In contrast, the iLQR optimality condition is inherently recursive: both the dynamics and cost functions depend on the optimal trajectory $x^*$ itself, yielding a relation of the form $x^* = f_{x^*}(x^*)$.

Such self-referential structure breaks the assumptions of standard auto-implicit differentiation, which cannot handle recursive dependencies without problem reformulation. Our method bridges this gap by recasting the iLQR recursion as a fixed-point equation suitable for exact differentiation—going beyond an implementation tweak to address a structural limitation in existing frameworks.

## 6.4. Limitations

Our method relies on the assumption that iLQR converges to a fixed point and requires access to both first-order and second-order derivatives of the dynamics. These constraints may limit its applicability to systems where such conditions hold.

# 7. Conclusions

In this work, we introduced DiLQR, an efficient framework for differentiating through iLQR using implicit differentiation. By providing an analytical solution, our method eliminates the overhead of iterative unrolling and achieves O(1) computational complexity in the backward pass, significantly improving scalability. Experiments demonstrate that DiLQR outperforms existing methods in both training loss and model loss, making it a promising approach for real-time learning-based control applications. In future work, we aim to explore relaxation techniques or alternative formulations to extend its applicability.

## Impact statement

This paper presents work aimed at advancing the field of Machine Learning. The societal implications of our research are discussed in the Discussion section, and we do not find any requiring additional emphasis here.

## Acknowledgments

We gratefully acknowledge the financial support of the Natural Sciences and Engineering Research Council of Canada (NSERC) and Honeywell Connected Plant. We also thank Dr. Brandon Amos, Dr. Nathan P. Lawrence, and the reviewers for their insightful discussions.

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

# A. Appendix

## A.1. Proof of proposition 1

**Proposition A.1.** *Define* $F_\theta := \frac{\partial F}{\partial \theta}$, $F_U := \frac{\partial F}{\partial U}$, $F_X := \frac{\partial F}{\partial X}$, $G_\theta := \frac{\partial G}{\partial \theta}$, $G_U := \frac{\partial G}{\partial U}$, $G_X := \frac{\partial G}{\partial X}$. *Define* $M := (I - F_X)^{-1}$, *and* $K := I - G_U$. *The analytical form of the gradients* $\frac{dX}{d\theta}$ *and* $\frac{dU}{d\theta}$ *are given as follows:*

$$\frac{dX}{d\theta} = M(F_\theta + F_U(K - G_X M F_U)^{-1}(G_X M F_\theta - G_\theta))$$
$$\frac{dU}{d\theta} = (K - G_X M F_U)^{-1}(G_X M F_\theta + G_\theta) \tag{18}$$

*Proof.* With the new notations, equations can be rewritten as:

$$(I - F_X)\frac{dX^*}{d\theta} - F_U\frac{dU^*}{d\theta} = F_\theta$$
$$-G_X\frac{dX^*}{d\theta} + (I - G_U)\frac{dU^*}{d\theta} = G_\theta \tag{19}$$

Focusing on the first equation, $\frac{dX}{d\theta}$ can be represented with $\frac{dU}{d\theta}$:

$$\frac{dX}{d\theta} = (I - F_X)^{-1}(F_\theta + F_U\frac{dU}{d\theta})$$
$$= M(F_\theta + F_U\frac{dU}{d\theta}) \tag{20}$$

Then, substituting 20 into the second equation of 19 to obtain an equation with respect to only $\frac{dU}{d\theta}$:

$$-G_X(M(F_\theta + F_U\frac{dU}{d\theta})) + (I - G_U)\frac{dU^*}{d\theta} = G_\theta \tag{21}$$

Solving equation 21 will give the solution to $\frac{dU^*}{d\theta}$:

$$\frac{dU}{d\theta} = (K - G_X M F_U)^{-1}(G_X M F_\theta + G_\theta) \tag{22}$$

Substituting 22 into 20, the solution to $\frac{dX}{d\theta}$ can be obtained:

$$\frac{dX}{d\theta} = M(F_\theta + F_U(K - G_X M F_U)^{-1}(G_X M F_\theta + G_\theta)) \tag{23}$$

This completes the proof. $\square$

## A.2. Experiments Details

We refer the methods in DiffMPC as DiffMPC.dx: Assumes the cost of the controller is known and approximates the parameters of the dynamics by directly optimizing the imitation loss; DiffMPC.cost: Assumes the dynamics of the controller are known and approximates the cost by directly optimizing the imitation loss. For all settings involving learning the dynamics (mpc.dx, DiffMPC.cost. DiLQR.dx, and DiLQR.cost.dx), a parameterized version of the true dynamics is used. In the pendulum domain, the parameters are the masses of the arm, length of the arm, and gravity; and in the cartpole domain, the parameters are the cart's mass, pole's mass, gravity, and length. For cost learning in DiffMPC.cost and DiLQR.cost, we parameterized the controller's cost as the weighted distance to a goal state $C(\tau) = \|w_g(\tau - \tau_g)\|$. As indicated in (Amos et al., 2018), simultaneously learning the weights $w_g$ and goal state $\tau_g$ was unstable. Thus, we alternated learning $w_g$ and $\tau_g$ independently every 10 epochs.

**Training and Evaluation** We collected a dataset of trajectories from an expert controller (iLQR with true system parameters) and varied the number of trajectories our models were trained on. The NN setting was optimized with Adam with a learning rate of $10^{-4}$, and all other settings were optimized with RMSprop with a learning rate of $10^{-2}$ and a decay term of 0.5.

## A.3. Detailed Network Architecture for Visual Control Task

**Encoder**    The encoder is a neural network designed to encode input image sequences into low-dimensional state representations. It is implemented as a subclass of `torch.nn.Module`, and consists of five convolutional layers and a regression layer:

- **Convolutional layers**: Each layer applies 2D convolutions, followed by batch normalization, ReLU activations, and max pooling. These operations progressively reduce the spatial dimensions of the input image.

- **Regression layer**: After the final convolutional layer, the output is flattened and passed through three fully connected layers, mapping the extracted features to the desired output dimension, which represents the system state.

The forward pass takes an input tensor of shape `[batch, 12, 224, 224]` (representing four stacked RGB images) and processes it through the convolutional layers. The output is a state vector of shape `[batch, out_dim]`.

**Decoder**    The decoder mirrors the structure of the encoder and is also a subclass of `torch.nn.Module`. It reconstructs images from the low-dimensional state vector. The decoder consists of five transposed convolutional layers followed by a regression layer:

- **Transposed convolutional layers**: These layers progressively upsample the input, applying batch normalization and ReLU activations after each layer to restore the spatial dimensions.

- **Regression layer**: This layer, consisting of three fully connected layers, transforms the low-dimensional input vector into a form suitable for the initial transposed convolution.

The forward pass takes a state vector of shape `[batch, 3]` as input, upscales it through the transposed convolution layers, and outputs a reconstructed image tensor of shape `[batch, 3, 224, 224]`. A Sigmoid activation is applied to ensure the pixel values remain within the range `[0, 1]`.

## A.4. Additional Experiments on Model Loss

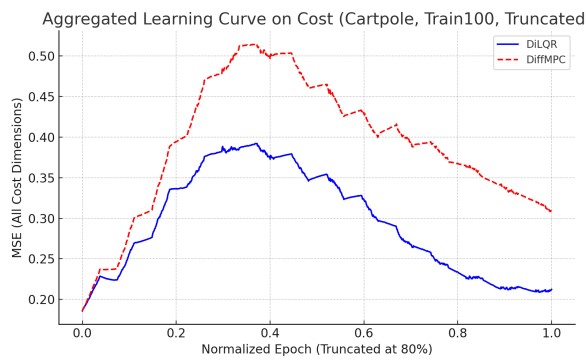

Figure 11: Aggregated Cost Learning Curve (Cartpole, Train=100, 500 epoch).

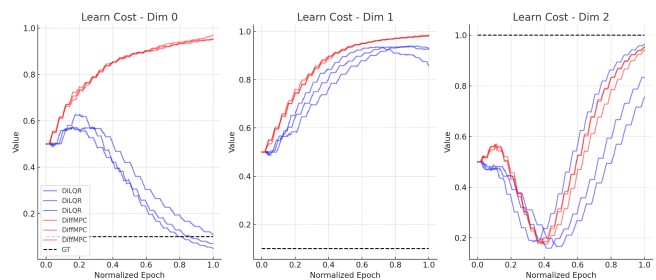

Figure 12: Per-Dimension Cost Learning Trajectories.

**Observation:**

DiLQR converges significantly faster than DiffMPC (Figure 11). In terms of dim-wise (Figure 12), although DiLQR and DiffMPC exhibit similar trends in Dim 1 and Dim 2, in Dim 0 the difference is significant: DiLQR (blue) steadily converges toward the ground truth (black dashed), while DiffMPC (red) consistently diverges in the wrong direction, highlighting a failure to capture the correct gradient signal.

## A.5. Note on Discrepancy from Rebuttal.

In our rebuttal, we reported a 0.0% bad-value ratio for DiLQR under train=100. That statistic was generated with the help of AI-assisted data summarization tool during fast-paced analysis, where truncated runs and inconsistent learning records were not fully filtered. In the final version, we reprocessed all experiments using a stricter pipeline, including run alignment, truncation exclusion, and ground-truth-based comparison. The resulting value (2.76%) remains significantly lower than DiffMPC (16.85%), and does not affect our original conclusions.

