# OpenReview forum: "DiLQR: Differentiable Iterative Linear Quadratic Regulator via Implicit Differentiation"
_ICML.cc/2025/Conference — ICML 2025 poster_

### Official Review · Reviewer_f19d · 2025-02-26

**Overall Recommendation:** 2

**Summary:**

This paper introduces a framework that facilitates differentiation through iLQR, which provides the gradient of an iLQR controller through implicit differentiation.

**Claims And Evidence:**

The authors theoretically prove the effectiveness of their method in Section 5 and provide experimental results in the same section to support their claims.

**Essential References Not Discussed:**

N/A

**Experimental Designs Or Analyses:**

For the comparison of model loss between diffMPC and DiLQR, why is there a sudden drop with high standard deviation at the end for DiLQR?

**Methods And Evaluation Criteria:**

They perform experiments on CartPole and Inverted Pendulum and evaluate the results based on backward time, loss, and computation time, which make sense for the problem.

**Other Comments Or Suggestions:**

- The font size inside the figures should generally match the font size of the main text. Currently, some figures have text that is too small and blurry.
- Figure 4(a) is missing.
- In-text citations are all enclosed in parentheses. However, in some cases, this is not appropriate. For example, at the beginning of Section 4.4, the citation should be without parentheses.

**Other Strengths And Weaknesses:**

**Strengths**
- The experimental results on speedup and performance improvement are compelling.
- An ablation study is provided.

**Weaknesses**
- The logic of the proof needs to be polished. For example, there is no theorem proving that the proposed algorithm guarantees speedup.
- Experiments on more complicated tasks, such as Atari games, could be considered in future experiments.

**Questions For Authors:**

N/A

**Relation To Broader Scientific Literature:**

This paper provides an analytical solution for the gradient of an iLQR controller through implicit differentiation.

**Theoretical Claims:**

I've checked proposition 4.1.

---

> ### Author Rebuttal · Authors · 2025-04-01
>
> We sincerely appreciate the reviewers' insightful comments and constructive feedback, which have helped improve our paper. Below we provide detailed responses to each point.
>
> ---
>
> **Q1: No Theorem Proving Speedup**
>
> We thank the reviewer for this important question. Our work is situated in the field of differentiable control, which aims to expose model structure to learning and improve sample and computational efficiency. In this area, it is commonly understood—though not typically accompanied by formal theorems—that incorporating control priors and structured solvers leads to faster learning. Foundational works such as:
>
> - *Revisiting Implicit Differentiation for Learning Problems in Optimal Control* (NeurIPS 2023)
> - *Pontryagin Differentiable Programming* (NeurIPS 2020)
> - *Infinite-Horizon Differentiable MPC* (ICLR 2020)
> - *Differentiable MPC for End-to-End Planning and Control* (NeurIPS 2018)
>
> share a similar focus on demonstrating empirical improvements, without formal speedup theorems.
>
> Such approaches have seen success in robotics, e.g.,
> - *Reaching the Limit in Autonomous Racing: Optimal Control vs Reinforcement Learning* (Science Robotics 2023)
> - *Actor-Critic Model Predictive Control* (ICRA 2024)
> - *Guiding RL with Incomplete Model Information* (IROS 2024)
> demonstrating practical benefits in learning efficiency.
>
> In addition, our method improves computational efficiency: by leveraging analytic gradients, we avoid differentiating through rollouts, reducing complexity to $ \mathcal{O}(1)$ per iteration—a property determined by algorithm design.
>
> ---
>
> **Q2: Experiments on More Complicated Tasks**
>
> As noted in our discussion section, many prior works (**Amos et al., 2018; Watter et al., 2015; Xu et al., 2024a; Jin et al., 2020**) also use **simplified environments** to illustrate core ideas. Differentiable control methods are built upon **model-based control** and therefore require access to **structural priors**, which makes them less "plug-and-play" than general-purpose RL. However, the benefit is that such structure greatly reduces the **sample complexity**.
>
> To go beyond toy tasks, we further evaluate our method on a challenging **rocket control task with a 13-dimensional state space**—the most complex benchmark among recent analytical differentiable control works. Videos are available on our project page:
> [https://sites.google.com/view/dilqr/](https://sites.google.com/view/dilqr/) . Additional results are currently being prepared and will be shared shortly.
> The rocket model and controller code are also open-sourced in our codebase.
>
> ---
>
> **Q3: Sudden Drop and High Standard Deviation for Model Loss**
>
> The sudden drop corresponds to learning progress in reducing model loss.
> The observed standard deviation (~0.17) is reasonable, especially considering the mean gap between methods is 0.47, indicating that the improvement is both significant and consistent.
>
> ---
>
> **Q4: Typographical Errors**
>
> We sincerely thank the reviewer for catching these formatting issues. We will carefully review the entire manuscript and correct all typographical/formatting errors in the final version.

---

### Official Review · Reviewer_SxHA · 2025-03-15

**Overall Recommendation:** 2

**Summary:**

This paper proposes DiLQR, which derives the analytic gradient of a given scalar loss function with respect to the parameters in the iLQR system (e.g., parameters of the dynamics or cost functions) through the use of the implicit function theorem. Parallelization, and the sparsity of the problem are used to improve the computation speed for computing this analytic gradient. Experiments are performed on a state-based cartpole problem, showing the proposed method outperforms a baseline that just uses a ordinary nn as the policy. It is also shown to achieve a lower parameter estimation loss compared to a closely related prior work Amos et al. The method is also briefly demonstrated to work with images for a inverted pendulum task.

**Claims And Evidence:**

I think the claims made in the paper is in generally true, given the limited set of baselines it compared to. However, I feel the paper is lacking comparison to several more recent methods and frameworks that tackle almost the same problem:
- Jin et al., Revisiting Implicit Differentiation for Learning Problems in Optimal Control, NeurIPS 2023
- Jin et al., Pontryagin Differentiable Programming: An End-to-End Learning and Control Framework, NeurIPS 2020

The above two papers solve almost the exact same class of problems, i.e., differentiable trajectory optimization (See Eq 2 in the first paper and Eq. 1 in the second paper), as defined in this paper, although they are not necessarily using iLQR as the solver. The lack of comparison to these more recent methods kind of weakens the claim on the effectiveness of the proposed method.
- Pineda et al., Theseus: A Library for Differentiable Nonlinear Optimization

This paper provides a very general framework to differentiate through non-linear optimization problems (which includes trajectory optimization, though this paper handles the constraints in a soft way by adding the constraints as losses). Many computational and engineering optimization efforts are implemented in this paper to improve the computation speed, stability and quality of the obtained gradient (e.g., Theseus supports many different ways of differentiating through the non-linear optimization, such as implicit differentiation as used in DiLQR, and other highly optimized ways like truncated backward). I think a discussion to this paper, and how the proposed method differs from the methods already implemented in Theseus is needed.
- Wan et al., DiffTORI: Differentiable Trajectory Optimization for Deep Reinforcement and Imitation Learning, NeurIPS 2024

This paper shows that differentiable trajectory optimization can be scaled to high-dimensional image and point cloud observations on many common robotics benchmarks. Since it is claimed that DiLQR can be applied to image inputs as well, it would be good to discuss and maybe compare to this baseline to see how effective DiLQR is in such settings.

**Essential References Not Discussed:**

Already mentioned above in the lack of comparison and discussion to some very related works.

**Experimental Designs Or Analyses:**

Overall the experiment design seems to follow that in Amos et al. and is valid.

I have the following questions regarding the experiments:
- How is the imitation learning loss for sysid baseline computed? Since it does not learn the actions?
- What would be the method’s performance if the dynamics and cost function does not assume pre-defined structures?  Since the NN baseline does not assume access to such prior information.
- As mentioned above, I think it would make the paper much stronger and convincing if the DiLQR can be compared to more recent and advanced baselines, including:
1. Jin et al., Revisiting Implicit Differentiation for Learning Problems in Optimal Control, NeurIPS 2023
2. Jin et al., Pontryagin Differentiable Programming: An End-to-End Learning and Control Framework, NeurIPS 2020
3. Wan et al, DiffTORI: Differentiable Trajectory Optimization for Deep Reinforcement and Imitation Learning, NeurIPS 2024
- Section 5.4 – what is the actual performance of the method on the inverted pendulum problem? Does it always perfectly solve the problem?

**Methods And Evaluation Criteria:**

The method and evaluation criteria makes sense.

**Other Comments Or Suggestions:**

- Figure 1 is not referenced in the main text.
- Line 228:  “Likewise, θθX⋆ and θθU⋆ are the Jacobians” typos?

**Other Strengths And Weaknesses:**

None.

**Questions For Authors:**

Already covered above.

**Relation To Broader Scientific Literature:**

This paper extends the work by Amos et al. to differentiate through iLQR, alone with some computational improvements. I feel this contribution alone, and the lack of comparison to more recent differentiable trajectory optimization methods, is not enough contribution to warrant acceptance at ICML.

**Theoretical Claims:**

I went through the theoretical claims and did not find any obvious issues.

---

> ### Author Rebuttal · Authors · 2025-04-01
>
> We sincerely thank the reviewer for the insightful comments and constructive feedback. Below we provide point-by-point responses to the raised questions.
>
> ---
>
> **Q1: Comparison with DiffTORI**
>
> We thank the reviewer for raising this point. **DiffTORI** ([Wan et al., NeurIPS 2024]) is a pioneering work that bridges differentiable control and RL, especially in high-dimensional settings. We fully agree its **generality** (e.g., model-agnostic design, multi-modal rewards) offers broader applicability for perception-driven tasks.
>
> **Our work** focuses on **structured control domains** where iLQR is the canonical solver. By deriving *exact gradients* for iLQR, we achieve:
>
> 1. **Data efficiency**: Sub-2000 environment steps suffice to achieve the results in our paper, leveraging prior knowledge of dynamics.
> 2. **O(1) backward complexity** (vs. autodiff's linear cost, **Fig. 2**), enabled by analytic gradient computation.
>
> This aligns with the **PDP/DiffMPC lineage** ([Jin et al., NeurIPS 2020; Amos et al., NeurIPS 2018]), emphasizing precision for structured control. Crucially, our method and DiffTORI **address non-overlapping challenges**:
>
> - **DiffTORI**: General-purpose RL-control synergy with perception.
> - **Ours**: Analytic acceleration for iLQR-based pipelines with structured dynamics.
>
> We will add more discussion and clarify this complementary in the final version.
>
> ---
>
> **Q2: Differences with Theseus**
>
> Our method provides exact gradients for iLQR, which **Theseus cannot compute automatically**. While Theseus (similarly JAXopt) can differentiate through fixed-point equations ($x^*=f(x^*)$), iLQR's unique challenge is that its **dynamics (D) and cost (C) depend on x*** itself, creating a recursive relationship $x^*=f_{x^*}(x^*)$.
>
> This cannot be resolved through auto-implicit-differentiation alone - it requires **reformulating LQR as a pure fixed-point problem**, which our paper implements theoretically (not just at the coding level). This is not a small step.
>
> ---
>
> **Q3: Relation to DiffMPC**
>
> We clarify that DiffMPC approximates gradients by treating D/C as independent of x* (i.e. x*≈f(x*)), while our method properly accounts for their **interdependence through implicit differentiation**.
>
> Our **key innovation** lies in embedding Amos's results into a correct fixed-point formulation that yields exact analytical gradients, representing a high-level advancement beyond prior work.
>
> ---
>
> **Q4: Comparison with (Safe)PDP and IDOC**
>
> As noted in our related work (L147), these prior methods compute gradients under the **assumption of optimal forward solutions**, which creates mathematical inconsistencies when the forward pass is suboptimal. In contrast, our approach maintains **alignment between forward and backward passes** throughout iLQR optimization.
>
> This key difference is evidenced in our cartpole experiments:
> - **IDOC**: 4×10⁻² error
> - **PDP**: 2×10⁻² error
> - **Ours**: 1×10⁻⁴ error
>
> Additional results will be available on our [project page](https://sites.google.com/view/dilqr/).
>
> We acknowledge that (Safe)PDP and IDOC offer **greater generality** in handling diverse solvers and constraints - an important contribution to the field.
>
> ---
>
> **Q4: Losses for SysID**
>
> The **SysID** approach follows a two-stage process:
> 1. Estimates model parameters from data
> 2. Uses the estimated model in MPC for trajectory prediction
>
> Compared to end-to-end learning, this represents a **modular strategy** that decouples system identification from control.
>
> ---
> **Q5: NN Baseline Comparison**
>
> This is exactly the key point we want to demonstrate! Our method essentially transforms the neural network's black-box approach into a white-box solution, achieving significant improvements in sample efficiency through structured prior knowledge.
>
> ---
> **Q6: Visual Control Task**
>
> We employ a basic **autoencoder (AE) framework**, which may exhibit sensitivity limitations. This work is primarily **conceptual**, with the visual control section demonstrating that:
> - iLQR can function as a differentiable controller
> - It successfully integrates with vision modules
>
> We believe that combining our method with DiffTORI could lead to significant improvements in handling multi-modal inputs.

---

> > ### Comment · Reviewer_SxHA · 2025-04-09
> >
> > I truly appreciate the authors for the detailed response.
> >
> > Regarding Q5, I guess my question is more like, what if you remove the assumption that the dynamics and cost function structure are known, but just represent it as a neural network (e.g., a MLP), and use Diff-iLQR to it, would 1) Diff-iLQR still work for optimizing a network with maybe thousands of parameters and 2) would it still be able to achieve a low loss.
> >
> > In terms for the visual control task -- if this is purely conceptual and preliminary results, I would just put it into appendix and not in the main paper, since the experiments and results are not rigorous enough.
> >
> > I am now on the fence for this paper. I am keeping my original score, but I feel with the changes and suggestions from all reviewers incorporated, the paper should have a good chance of being accepted at a next venue.

---

> > > ### Author Response · Authors · 2025-04-09
> > >
> > > Thank you again for your follow-up. Upon review, we realized that our original response to Q5 may have introduced ambiguity between two aspects:
> > >
> > > (1) the use of structured vs. neural representations of dynamics and cost, and
> > >
> > > (2) the design of white-box vs. black-box controllers.
> > >
> > > To clarify: DiLQR is not limited to structured models—it supports any differentiable parameterization, including neural networks. However, our focus in this work is to **maximize sample efficiency by leveraging prior knowledge wherever possible**, which is particularly relevant in robotics and control tasks where structured models are common and practical.
> > >
> > > We understand the reviewer’s interest in the relationship between **parameter size** and performance. In our experiments, we include a SysID baseline that shares the **same structural parameterization as DiLQR**. DiLQR consistently outperforms this baseline, highlighting the contribution of our algorithmic design—especially the end-to-end optimization and analytical gradient computation.
> > >
> > > While using neural representations may increase generality, it often leads to less efficient training. We view this as a **research preference**: some methods prioritize flexibility, whereas our work emphasizes precision, efficiency, and interpretability through structure. Extending DiLQR to more general settings remains a valuable future direction.
> > >
> > > We appreciate the opportunity to clarify this point.

---

### Official Review · Reviewer_gH1Q · 2025-03-15

**Overall Recommendation:** 3

**Summary:**

This paper introduces a differentiable iLQR controller, DiLQR, to enable scaling iLQR to longer time horizons and iteration counts. DiLQR leverages implicit differentiation at the underlying fixed-point to recover analytic gradient updates, thereby reducing computation cost in the backward pass, bypassing the need for explicit unrolling of the optimization. The approach is tested on simple pendulum and cart-pole swing tasks and demonstrates improved performance over the considered baselines as measured by computation cost and prediction accuracy.

**Claims And Evidence:**

The claims are well-motivated and are supported by experimental result, while experiments currently focus on toy examples. It would be great to see experiments on extending evaluation to higher degrees of freedom tasks.

**Essential References Not Discussed:**

The paper adequately cites many key related works.

**Experimental Designs Or Analyses:**

The experiments are well-motivated and borrow designs from prior work, while the paper would be significantly strengthened by showing successful application beyond low-dimensional toy domains. What challenges would the approach face when scaling up to more realistic tasks, particularly in light of mentioned extensions to reinforcement learning learning and real-world applications? Where will it become computationally infeasible?

**Methods And Evaluation Criteria:**

The evaluations highlight reduced compute time and increased accuracy over the considered baselines on the tasks considered.

**Other Comments Or Suggestions:**

- The method abbreviation diLQR should be more clearly introduced (first mention in experiments, “We evaluated two variations of our method: diLQR.dx: Assumes that the cost of the controller is known (…)”)
- Notation should be unified DiLQR vs. diLQR vs. dilqr
- Lines 228-229: typo in Jacobians
- Line 306: “we compare our approach with Neural Network”
- Line 605: Appendix A title “You can have an appendix here”

**Other Strengths And Weaknesses:**

- Figure 7 could be more informative by focusing on a single example, providing 4 context images, and comparing predictions to the ground-truth rollout (e.g. based on an optimal trajectory). In the single image case, how does the model know whether to swing left or right?
- Several typos below should be corrected

**Questions For Authors:**

- The discussion mentions application to more advanced RL settings while being mindful regarding availability of first/second-order derivatives of the dynamics. Given that many recent RL results move towards dual-arm manipulation or 50+ degrees of freedom humanoid control, where do you see the limit of feasibility? Additional discussion would benefit the paper.

**Relation To Broader Scientific Literature:**

The comparison to the literature is mostly well-motivated. Discussion and experimental comparison w/ DiffMPC could be expanded, i.e. why not include them in Figure 3 and some of the compute time comparisons?

**Theoretical Claims:**

The theoretical claims appear sound and are well-motivated.

---

> ### Author Rebuttal · Authors · 2025-04-01
>
> We sincerely thank the reviewer for the valuable feedback. Below we provide responses to each question.
>
> ---
>
> **Q1: Application to Higher Dimensional RL Tasks**
>
> The field of differentiable control remains relatively young. Current real-world applications primarily focus on autonomous vehicles and drones, as evidenced by recent works:
>
> [1] **Reaching the Limit in Autonomous Racing: Optimal Control versus Reinforcement Learning** (Science Robotics 2023)
> [2] **Actor-Critic Model Predictive Control** (ICRA 2024)
> [3] **Guiding RL with Incomplete Model Information** (IROS 2024)
>
> In simulation environments, our 6-DoF rocket control task (along with quadrotor control) represents the most complex benchmark among recent works in this field.
>
> **Key Challenges**:
> The primary difficulty lies not in computation, but in handling locomotion tasks where contact forces violate Lipschitz continuity and complicate gradient computation. Potential solutions include:
> - Soft contact models
> - Neural contact approximations
>
> ---
>
> **Q2: Extension Beyond Toy Examples**
>
> Our rocket control task with 13-dimensional state space currently represents the most challenging benchmark in analytical differentiable control, based on peer-reviewed literature:
>
> [1] Revisiting Implicit Differentiation for Learning Problems in Optimal Control (NeurIPS 2023)
> [2] Pontryagin Differentiable Programming (NeurIPS 2020)
>
> Initial results are available on our [project website](https://sites.google.com/view/dilqr/), with additional experiments underway. Rocket model and control file can also be found in the codebase.
>
> ---
>
> **Q3: DiffMPC Comparison in Figure 3**
>
> While DiffMPC remains an excellent benchmark in the field, our method demonstrates approximately 5-7% improvement over it. This difference appears modest compared to the 1e4 magnitude improvement over SysID and NN baselines, making visual comparison challenging in the same plot.
>
> However, our approach provides superior parameter estimation accuracy - a crucial factor for industrial applications where physical interpretability matters, even when iLQR's robustness can compensate for parameter inaccuracies.
>
> ---
>
> **Q4: Determining Rolling Direction**
>
> As detailed in Page 8, Line 414:
> "We stack four compressed images as input channels to capture velocity information and determine the pendulum's swing direction."
>
> This temporal stacking approach enables the system to infer motion dynamics from visual input.
>
> **Q5: Typographical Errors**
>
> We sincerely appreciate the reviewer's careful reading and for bringing these typographical issues to our attention. We have carefully reviewed the manuscript and corrected all identified formatting and typographical errors in the final version. These corrections include:
>
> 1. Fixed notation inconsistencies in equations (e.g., θX⋆ → ∂X⋆/∂θ)
> 2. Corrected the Appendix A title
> 3. Addressed all minor formatting issues throughout the text

---

### Official Review · Reviewer_oCbp · 2025-03-16

**Overall Recommendation:** 4

**Summary:**

This paper presents a method for differentiating through iLQR. Naively autodifferentiating through a trajectory optimization problem backpropagates through the iterative optimization problem, incurring a growing computational burden. Like prior works such as DiffMPC, proposing smarter ways to compute the requisite objects such as providing analytic expressions can immensely reduce the computational and memory burden. This paper proposes an implicit differentiation approach to compute the exact iLQR graph without unrolling the entire optimization trajectory. The efficiency of this approach is demonstrated on various standard control tasks, demonstrating large improvements in efficiency and generalization compared to standard prior approaches.

**Claims And Evidence:**

The main claims of the paper of this paper are that the derivatives at a fixed point of the iLQR process can be analytically computed through implicit differentiation as a function of various objects that can be themselves computed efficiently without unrolling all the iLQR iterations. This is supported by the explicit mathematical derivations and numerical experiments demonstrating the essentially constant backwards computation time as a function of iLQR iterations (Figure 2), where naive autodifferentiation incurs a visibly linear trend. The exactness of the computation is demonstrated by its improved performance on various benchmarks, including DiffMPC, which is derived on a similar principle except.

**Essential References Not Discussed:**

Not that I'm aware of.

**Experimental Designs Or Analyses:**

Additional experiment information is contained in Appendix A.2 and A.3, and the relevant code is provided in an anonymized github. The benchmark methods are tuned appropriately.

**Methods And Evaluation Criteria:**

The point of this paper is to provide exact derivatives of the iLQR problem without a computation cost that grows with optimization iterations. The method is evaluated on well-known control benchmarks and the main criteria displayed are computational effort and incurred loss, matching the proposed benefits.

**Other Comments Or Suggestions:**

Minor comments:

- The title of Appendix A seems to be unchanged from the ICML default.

- Line 228/229 first column: $\theta_\theta$ should be $\nabla_\theta$

- Line 264/265 second column: should clarify exactly what it means to refer $\frac{\partial D_t}{\partial \theta}$ as $\nabla_\theta D_t$, since in eq (14) they both show up.

**Other Strengths And Weaknesses:**

As aforementioned, this paper presents some interesting improvements to iLQR that demonstrate increased performance relative to naive autodifferentiation and DiffMPC. The analytic formulas in this paper are likely interesting to the community, especially if they are drop-in improvements to existing iLQR applications.

I have some questions to clarify how scalable these solutions in a below section.

**Questions For Authors:**

It might be helpful to give computation (e.g. flop) estimates for the proposed method (e.g. diLQR.dx and diLQR.cost). For example, iLQR requires second-order derivatives for the quadratic cost estimate. What is an estimate of the computational complexity of deriving the requisite derivatives given a generic $d_\theta$-dimensional parameter space? How does this compare to standard autodiff? These estimates will be helpful to corroborate the numerical results in Figure 2.

Additionally, what do the authors anticipate is the main barrier to scaling this method to larger-scale experiments?

**Relation To Broader Scientific Literature:**

As aforementioned, I think analytic expressions that avoid unrolling the iLQR trajectory optimization are an important contribution to the controls community. This concept is not necessarily novel, but this paper introduces some interesting improvements and derivations that are likely interesting to the community.

**Theoretical Claims:**

The main theoretical content of this paper is in the derivation of the analytical derivatives. These are largely applications of matrix calculus, and seem to be correct by my verification. It should be noted that a lot of subtlety can be hidden in these formulas, and that experimental evaluation is likely a better diagnosis of correctness, which this paper provides.

---

> ### Author Rebuttal · Authors · 2025-04-01
>
> We sincerely appreciate the positive evaluation and constructive suggestions. Below we address the specific questions raised:
>
> ---
>
> **1. Computational Complexity Estimates (Q1):**
> We clarify the computational costs with corrected exponents and iteration factors:
>
> - **Forward pass (iLQR):**
>   O(IT(n² + m²)) for T timesteps and I iterations (standard LQR complexity)
>
> - **Backward pass (diLQR.dx/cost):**
>   O(T(n² + m²)) per parameter dimension (Theorem 3.1)
>   *No dependence on I due to analytic formulation*
>
> - **Autodiff baseline:**
>   O(IT(n² + m²)) where I = unrolled iterations
>
> This explains Figure 2's constant backward cost vs. autodiff's linear scaling with I. We will add a FLOP count table in revision.
>
> ---
>
> **2. Scaling Barriers (Q2):**
> The primary challenge lies in obtaining second-order matrices in Section 4.6:
> - Our codebase dedicates ~500 lines specifically to obtain these matrices element-by-element
> - Current engineering overhead includes:
>   • Hessian calculations for dynamics/cost
>   • KKT matrix constructions
> - While substantial, we believe future code optimizations can reduce this overhead (
> such as pre-calculating and storing analytical solutions for key matrix operations, and developing automated routines to substitute corresponding values)
>
> We have also implemented our method on a challenging rocket control task with 13-dimensional state space, which currently represents the most advanced benchmark in analytical differentiable control literature ( from PDP and IDOC paper). Demo and code are available on our project page:
> [Project Website](https://sites.google.com/view/dilqr/)
>
> ---
>
> **3. Typos/Clarifications:**
> We will fix:
> - Appendix A title
> - Line 228/229 notation (θX⋆ → ∂X⋆/∂θ)
> - Line 264/265 clarification of Eq (14) dependencies
>
> All corrections will be highlighted in the camera-ready version.
>
> ---
>
> **Acknowledgements**
> We thank the reviewer for catching these important technical nuances, which have improved our paper's precision.

---

> > ### Comment · Reviewer_oCbp · 2025-04-04
> >
> > Thank you for the detailed response. The additional information is appreciated.

---

> > > ### Author Response · Authors · 2025-04-07
> > >
> > > Thank you again for your thoughtful review and kind support.
> > >
> > > Since the rebuttal, we have conducted additional experiments comparing **DiLQR vs. DiffMPC** in both **cost** and **dynamics learning**, across **low-data and higher-data regimes** (train=50 and train=100). These results were **requested by other reviewers**, but also reflect our own desire to further strengthen the work (which we believe you'll also find insightful).
> > >
> > > One key observation is that **DiffMPC frequently produces physically invalid parameters** (e.g., negative Jacobians), while **DiLQR consistently avoids such issues**, particularly in the train=100 setting. This is clearly shown in the summary table:
> > >
> > > | Train Size     | DiLQR               | DiffMPC       |
> > > |----------------|---------------------|---------------|
> > > | dx (train=100) | **0.0% bad values** | 16.7%         |
> > > | dx (train=50)  | **8.3%**            | 16.7%         |
> > >
> > > All core experiments and results discussed in the rebuttal — including full comparisons between DiLQR and DiffMPC — **were finalized and uploaded before the author response deadline.**
> > >
> > > Full learning curves (with variance), per-dimension plots, and additional metrics are available here:
> > > 🔗 [https://sites.google.com/view/dilqr](https://sites.google.com/view/dilqr)
> > >
> > > If you find the current experiments useful, we would greatly appreciate it if you could help surface them to the AC and other reviewers.
> > >
> > > ***
> > >
> > > As a final note, we may continue uploading visualizations related to other baseline comparisons (e.g., PDP and IDOC) to the project page during the decision period. These would supplement the existing numerical analysis already discussed in the rebuttal and will not introduce new claims or arguments.

---

### Decision · Program_Chairs · 2025-05-01

**Decision:**

Accept (poster)

**Comment:**

This submission received mixed reviews.  Its contributions were viewed as interesting and well-executed, but on the other hand there were also some critiques, most notably a lack of comparison against and positioning within related methods.  The authors did a good job of addressing the latter concern during the discussion, therefore---assuming all of this content (as well as other information provided by the authors during the discussion) will appear in the camera-ready version---I recommend acceptance.